# Numerical Ecology and Social Network Analysis of the Forest Community in the Lienhuachih Area of Taiwan

Tung-Yu Hsieh [1,2,3,4,*], Chun-Jheng Yang [5], Feng Li [2,3,4] and Chyi-Rong Chiou [5,*]

1   School of Food and Bioengineering, Fujian Polytechnic Normal University, Fuqing 350300, China
2   Fujian Universities and Colleges Engineering Research Center of Modern Facility Agriculture, Fujian Polytechnic Normal University, Fuqing 350300, China
3   Lanxi Agricultural Technology Co., Ltd., Fuqing 350300, China
4   Wuyou Ecological Agriculture Co., Ltd., Fuqing 350300, China
5   School of Forestry and Resource Conservation, National Taiwan University, No. 1, Sec. 4, Roosevelt Rd., Taipei 10617, Taiwan
*   Correspondence: dyxie@sibs.ac.cn (T.-Y.H.); esclove@ntu.edu.tw (C.-R.C.)

**Abstract:** In this study, the integration of useful statistical methods from different disciplines for analyzing the forest community of the Lienhuachih area of central Taiwan was attempted. We employed a seriated heat map to confirm the presence of multiple community patterns in the area and used the gap statistics and a clustplot to confirm the number and structure of the patterns, respectively. A minimum spanning tree was used to display a succession series among the quadrats, and Renyi diversity was used to indicate the species composition of these patterns. The results confirmed the existence of six patterns with different biodiversity structures in which pattern C was the succession prototype of the local community patterns. Next, we used the patterns as nodes of a social network to perform bipartite network analysis. The results showed that a community network consisted of 108 taxa and six syntaxa. The syntaxa were highly vulnerable to extinction; therefore, the optimal conservation strategy for local species would be to first protect the syntaxa. The random forest method and bipartite modularity were used to analyze the dominant characteristic species of the six syntaxa. The results showed that these two methods are useful for detecting characteristic species of the syntaxa. Therefore, this study proposed a new nomenclature system, namely the Mathematic Code of Syntaxonomic Nomenclature, to support the results of numerical vegetation analysis. Finally, the potential for an apparently competitive network was examined, the role of an apparently competitive network in the local structuring community was explored, and six new associations in the Lienhuachih area were described.

**Keywords:** numerical taxonomy; random forest; bipartite networks; modularity analysis; apparent competition; minimum spanning tree; Renyi diversity; new syntaxa; CART; MCSN; taxonomy modeling; classifier modeling

## 1. Introduction

Humboldt, who first proposed the basic concepts of vegetation appearance and physiognomy, originally propounded vegetation ecology, also known as plant community ecology [1]. However, the titles, concepts, definitions, methods, and focus areas of early studies on vegetation ecology varied considerably; consequently, multiple terms, such as phytocoenology, symmorphology, synecology, symphysiology, syntaxonomy, and syndynamics, were used to indicate vegetation ecology. Since the early 19th century, some ecologists have studied plots of vegetation, which they considered samples of a plant community to study as integrated units which can be classified, described, and analyzed. Early vegetation classification efforts were driven largely by a desire to understand the natural diversity of vegetation and the factors that create and sustain it. Vegetation classification is critical to basic scientific research as a tool for organizing and interpreting information and

placing that information in context. The use of vegetation classification has increased over the past few decades. Vegetation description and classification provide units critical for the inventory and monitoring of natural communities, planning and managing conservation programs, documenting the requirements of individual species, monitoring the use of natural resources such as forest and range lands, and providing targets for restoration. Vegetation types are even achieving legal status where they are used to define endangered habitats and where their protection is mandated [2].

By far the most widely applied approach to vegetation classification is that developed by Josias Braun-Blanquet [3]. The method centers on recording the species composition of the basic unit (plot or relevé) and the association as a vegetation unit that could be classified in a similar way to a species. Associated with the plot are records of its location, size, physical setting, and vegetation composition. The advantages of the Braun-Blanquet system include the consistency of the approach, the enormous number of plots that have been recorded, and a large number of published descriptions of vegetation types. Weaknesses include a seemingly arbitrary definition of units, the lack of a requirement that new units be integrated with established units, and the lack of any formal registry of published units [2]. After a long period of continuous development, vegetation ecology became a sub-discipline of modern ecology and now mainly involves studies on the species composition, function, dynamics, succession, classification, and distribution of plant communities [4,5].

Early classification of vegetation was subjective. Numerical methods were developed to provide objective procedures. Hence, the most common approach nowadays to vegetation classification is by numerical means [4]. Typically, this requires defining a similarity or dissimilarity matrix among all of the vegetation plots and then clustering the plots into types. Numerous methods of classification were developed with various similarity measures and different strategies for grouping plots together [6]. These methods of numerical ecology were introduced from the field of numerical taxonomy at the end of the past century and have been recently used for advancements in the study of vegetation ecology [5,7]. With the availability of textbooks and software packages, there has been a massive expansion in the use of numerical methods in vegetation science and other areas of ecology [6,7]. The development of numerical methods in vegetation ecology has led to considerable advancements in the discipline of analogy. Similarly, research in the social sciences has been supported by social network analysis in the past decades [8]. However, the methods of social network analysis are rarely applied to vegetation ecology. Interdisciplinary studies are of high importance and are needed to solve the complex problems of vegetation classification. Progress in vegetation science depends on the development of explicit theory and numerical methods capable of discriminating between rival theories [4]. Therefore, in the present study, we used the plots in the Lienhuachih area of central Taiwan as a case study, attempted to synthesize methods originating from two disciplines with similar theories, and explored the ecological implications of the results obtained by combining these methods. We also hope that this will help vegetation ecologists progressively improve the statistical methods used in vegetation ecology through this early interdisciplinary work.

## 2. Materials and Methods

### 2.1. Study Areas

Because previous related studies [9–12] have provided extremely detailed information about the Lienhuachih area, herein we provide only a summary of relevant information. The study plots of Lienhuachih were set up in 2008. The geographical location is approximately 23°54′49″ N and 120°52′43″ E; the elevation ranges from 667 to 845 m, and the average slope is 35.3°. The mean annual temperature is 20.8 °C; the rainy season in this region lies between May and September, and the annual precipitation is approximately 2285 mm in the Lienhuachih area.

*2.2. Experimental Design*

The vegetation survey in the Lienhuachih area was conducted in the year 2012. Abundance was recorded as the number of tree species, and sampling and data collection were conducted according to the standardized protocols adopted by the Center for Tropical Forest Science plots network. In total, 25 quadrats, each measuring $20 \times 20$ m$^2$, were systematically sampled within the study site. Each quadrat was spaced at 100 m and was arranged in a square grid with a projected area of 25 ha of evergreen forest. The codes used to identify the quadrats and species were according to those used in a previous study [11]. The species code accompanies the scientific name at the first mention in this article.

*2.3. Statistical Methods*

Multiple methods have been applied to numerical taxonomy/ecology and social network analysis in recent years [5,7,8,13]. In this study, we carefully applied methods from different disciplines. All data analysis and statistical methods in this study were conducted using R software packages [14] according to the following steps:

Vegetation classification:

1. The seriated heat map is a novel method that has been recently applied to numerical taxonomy [13]. The advantage of this method is that it simultaneously implements clustering and ordination visualizations in a plot [15,16]. The Q-Q-type seriated heat map [13,15,16], which employs the Bray–Curtis dissimilarity index [17–19], was used to detect possible community patterns in this study.
2. The gap statistic, Gapk, is a goodness of clustering measure obtained after running 10,000 Monte Carlo samples [20,21] and was employed to estimate the number of community patterns hidden in the seriated heat map. Next, the detected community patterns were coded in capital letters for subsequent analyses.
3. We used the results from the seriated heat map and gap statistics and applied the clustplot by using the partitioning around medoids (PAM) algorithm [20,22,23] to display a clustering ordination of the community patterns.
4. A minimum spanning tree was embedded into a PCoA ordination [5,18]; this was used to determine the shortest path of succession among the quadrats.
5. Renyi diversity [24] profiles were used to determine the biodiversity structure of the community patterns [7,18,25–28].

Social network analysis:

1. The community patterns and their species were used as nodes in a social network; they were then converted into a taxon–syntaxon bipartite web for social network analysis [29–32]. All of the selected indices were calculated [33] to describe network properties [34,35] in this study.
2. The characteristic value of a species (IncNodePurity) was determined using the random forest model for each community pattern consisting of 100,000 trees. The characteristic species of each syntaxon was determined [36–39].
3. We conducted bipartite modularity by applying Newman's modularity measure in a weighted web to detect the modules and compute their information in the bipartite network [33,40–42].
4. The potential for apparent competition (PAC) among or within community patterns [43–45] was calculated using a previously reported formula [46–49].
5. This study adopted the classification and regression tree [50] of the dominant/characteristic species as the key to syntaxa in the Lienhuachih area of Taiwan.

## 3. Results

*3.1. Vegetation Classification:*

Figure 1 is the Q-Q type seriated heat map of the 25 quadrats used in this study; the map appears to have several inconspicuous pattern blocks, but the blocks are vague. Therefore, we calculated the Gapk [20,21] to confirm the number and validity of the patterns

in this study. The result (Figure 2) showed that a steep gap appeared when the samples were divided into 6 or 23 clusters. Based on the fact that, only between 5–6 or 22–23 clusters, there is a bigger difference in the average of clusters than the standard deviations of 5–6 or 22–23 clusters, dividing the samples into 6 or 23 clusters was reasonable. However, when the samples were divided into 23 clusters, the clusters were very similar to the individual quadrats, which would not be useful for pattern detection. Hence, the 25 quadrats were divided into 6 clusters, which was the only reasonable pattern structure apart from the individual structure. The two results, Figures 1 and 2, consistently indicated that six community patterns were hidden in the 25 squares. The detected community patterns were then randomly coded using capital letters (from A to F).

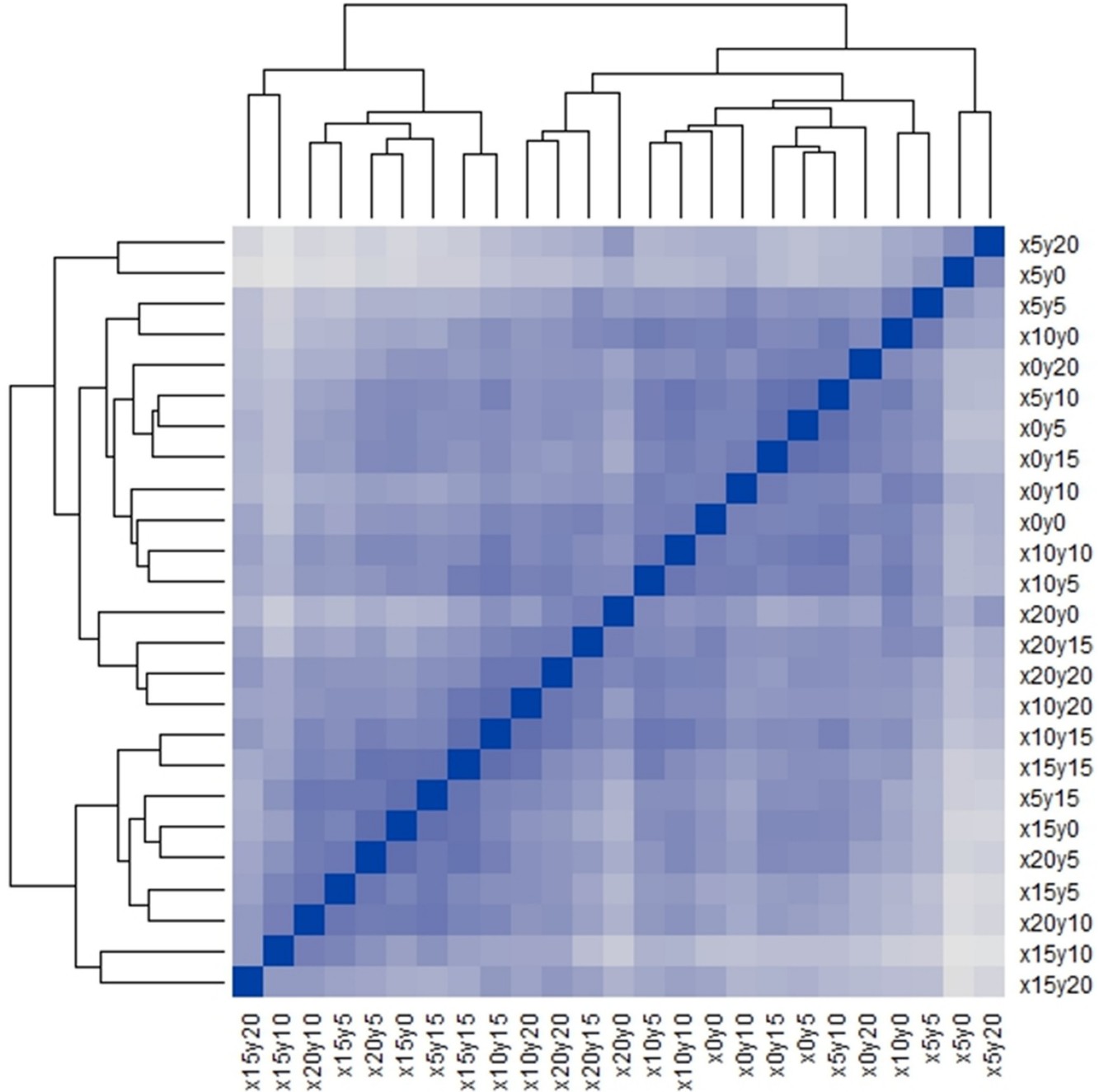

**Figure 1.** Seriated heat map of 25 quadrats. The blue-scaled color as a meter indicates the value of similarity.

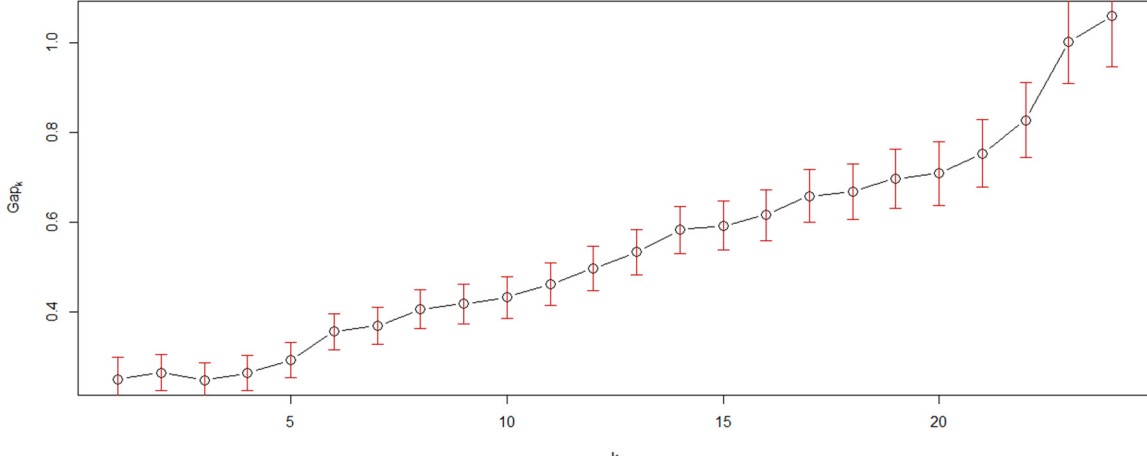

**Figure 2.** $Gap_k$ and its standard deviations came from 10,000 Monte Carlo samples that were calculated to estimate the number of clusters k, k from 1 to 24.

To reveal the clear structure of hidden patterns, a clustplot (bivariate cluster plot), made using the PAM algorithm [20,22,23] was used to visualize the distinct community patterns as well as the community members. The result showed that the forest community of the study site could be divided into six patterns, and an ellipse was drawn around each pattern to indicate its members. The relationships among the patterns were indicated using red lines (Figure 3). The quadrats x15y20, x15y5, x20y20, x5y10, x10y0, and x5y20 were the six medoids of patterns A–F, respectively. In the next step, a minimum spanning tree embedded into a PCoA ordination [5,18] was conducted to determine the shortest path of succession among the quadrats. The graph shows the successional series in the form of a tree. Figure 4 shows that the center and five main branches of the minimum spanning tree correspond to the six patterns in Figure 3. The two aforementioned methods were complementary. Figure 3 reveals the relationship between the patterns, and Figure 4 reveals the relationship between the members of the patterns. Combining the results of these two methods, the internal and external structures of the community patterns were completely revealed. Figures 3 and 4 show that C is the center pattern in the forest community of the study site. A is a unique pattern that contains only one quadrat (x15y20).

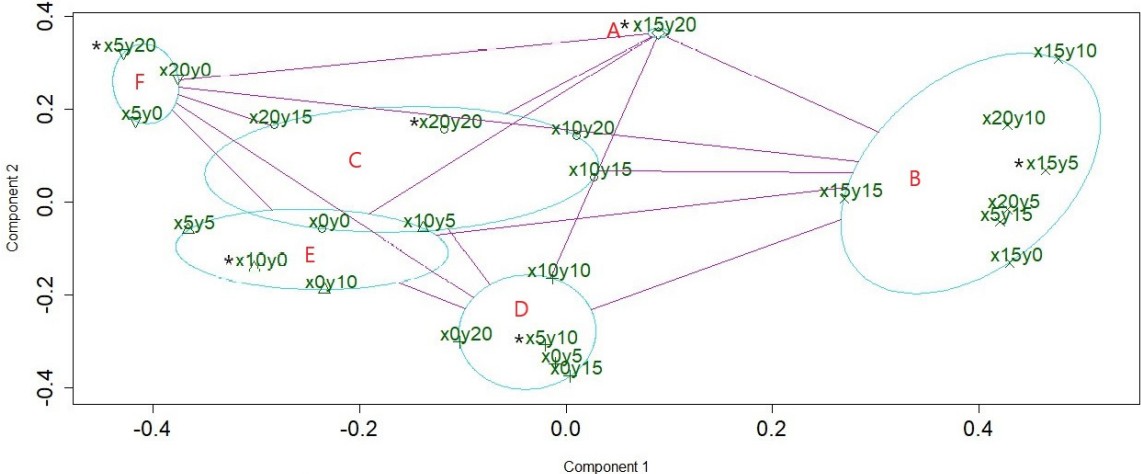

**Figure 3.** The clustplot shows the relationship of six community patterns with their members. The first two principal components explained 43.21% of the point variability. The medoid of a cluster is indicated with a star.

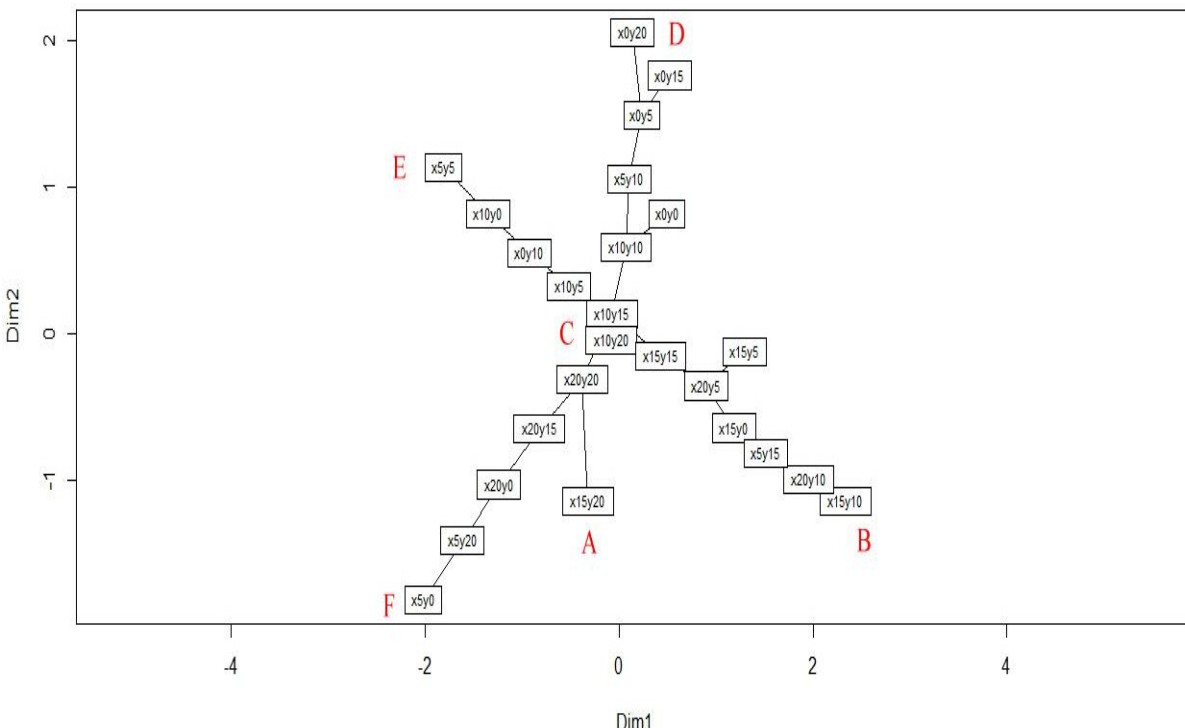

**Figure 4.** The minimum spanning tree of succession series from 25 quadrats in the forest community of Lienhuachih area.

Renyi entropy [24] was adopted to reveal and compare the biodiversity structures of these community patterns by using the Renyi diversity profiles [7,18,25–28]. The result (Figure 5) demonstrated that biodiversity structures differed among the six patterns; D exhibited the highest diversity along the upper line, and A exhibited the lowest diversity along the lower line. The median lines of Renyi diversity (red lines) in Figure 5 show that patterns B and E have relatively high species richness (scale = 0), but the Shannon (scale = 1) and Simpson (scale = 2) diversities of patterns B and E are similar to the median lines. The species richness of pattern E is higher than that of F, but the Shannon, Simpson, and Berger–Parker (scale = ∞/Inf) diversities of these two patterns are similar and along the median lines. The ordering, in terms of Berger–Parker diversity, of these patterns in descending order is D, C, F, E, B, and A.

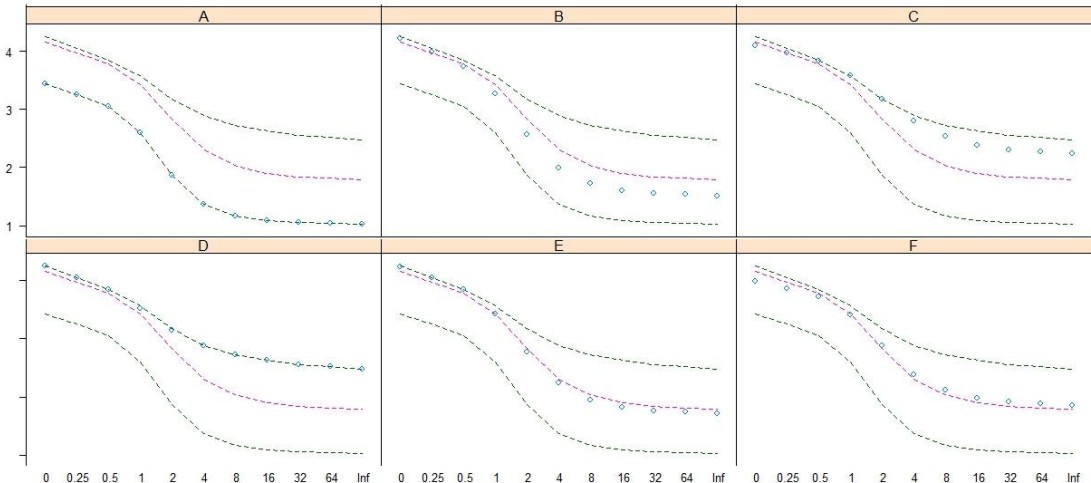

**Figure 5.** The Renyi diversity profiles show the biodiversity structure of patterns (**A**–**F**). The *y*-axis and *x*-axis are the values and scales of Renyi diversity, respectively.

*3.2. Social Networks Analysis*

We used the taxa and syntaxa as nodes in social network analysis and created a taxa–syntaxa bipartite ecological web for social network analysis [29–32]. Hence, the tree species were the higher taxa, whereas the patterns were the lower syntaxa in the bipartite web. Multiple indices can be used to describe the network properties or topography, but some of them do not apply to ecological networks [32]. Hence, listing all of the indices for the bipartite network in this study is unnecessary. We selected some useful indices to reveal network properties and their ecological implications in this study. The results (Table 1) revealed that the web size was 108 taxa and six syntaxa. The connectance index is the standardized number of species combinations often used in co-occurrence analyses [49]. A connectance index of 0.54 indicated that approximately half the tree species in a web could coexist in the same syntaxon. If the number of compartments was one, it indicated that the web was not separated into sub-sets [51]. A web asymmetry of 0.89 indicated that the number of taxa was considerably higher than that of syntaxa [52]. The extinction slopes for the taxa and syntaxa were 77.41 and 3.52, respectively. That extinction slope values indicated that the syntaxa were more vulnerable than the taxa because of high slope estimates, indicating relatively few effects of extinction on the network [35]. The C-scores of the taxa and syntaxa were 0.38 and 0.2, respectively, which indicated that the syntaxa were more aggregated than the taxa [53]. The V-ratios showed that the syntaxa and taxa exhibited positive aggregations [54]. High values of the togetherness of the syntaxa indicated that the taxa exhibited more disaggregation than the syntaxa [53]. The mean number of shared syntaxa per taxon indicated that an average of 1.85 syntaxa was shared by a taxon. The mean number of shared taxa per syntaxon indicated that an average of 37.27 tree species was recorded in a syntaxon [53]. The values of robustness indicated that the syntaxa were more vulnerable to secondary extinction [35].

**Table 1.** All selected indices for the taxon-syntaxon bipartite network of study site.

| Indices | Values |
| --- | --- |
| number of syntaxa | 6 |
| number of taxa | 108 |
| connectance | 0.54 |
| number of compartments | 1 |
| web asymmetry | 0.89 |
| extinction slopes of taxon | 77.41 |
| extinction slopes of syntaxon | 3.52 |
| C-Score of taxon | 0.38 |
| C-Score of syntaxon | 0.2 |
| V-Ratio of taxon | 2.14 |
| V-Ratio of syntaxon | 10.39 |
| togetherness of taxa | 0.17 |
| togetherness of syntaxa | 0.35 |
| mean number of shared syntaxon per taxon | 1.85 |
| mean number of shared taxa per syntaxon | 37.27 |
| robustness of taxa | 0.99 |
| robustness of syntaxa | 0.76 |

A random forest with 100,000 trees was used to measure the characteristic values of species for each syntaxon. Table 2 lists the first five species with high characteristic values for each syntaxon. The characteristic species of syntaxon A are GORDAXP (*Gordonia axillaris* (Roxb.) Dietr.), MELACA (*Melastoma candidum* D. Don), and ILEXLO (*Ilex lonicerifolia* Hayata), with a characteristic value of approximately 0.16. Thus, this community exhibited a high abundance of GORDAX, ILEXLO, and MELACA. The number of GORDAX in this syntaxon was as high as 57, but it did not exceed 12 in the other syntaxa, while MELACA in this syntaxon was 7, but it did not exceed 3 in the other syntaxa. The characteristic species of syntaxon B were ILEXGO (*Ilex goshiensis* Hayata), SYZYBU (*Syzygium buxifolium*

Hook. and Arn.), and RANDCO (*Randia cochinchinensis* (Lour.) Merr.). They were rare in the other syntaxa, but syntaxon B was rich in these species. The characteristic species of syntaxon C was MELISQ (*Meliosma squamulata* Hance), with a characteristic value of 0.23. The characteristic species for syntaxa D, E, and F were PSYCRU (*Psychotria rubra* (Lour.) Poir.), HELIFO (*Helicia formosana* Hemsl.), and CINNSU (*Cinnamomum subavenium* Miq.), respectively. Notably, in syntaxon F, CINNSU and ORMOFO (*Ormosia formosana* Kanehira) were negatively characteristic species for F; therefore, CINNSU and ORMOFO existed in all syntaxa except for F. In this case, we only employed the third-order species, FICUFI (*Ficus fistulosa* Reinw. ex Bl.) or SAURTR (*Saurauia tristyla* DC. var. *oldhamii* (Hemsl.) Finet and Gagnep.), but these two species had low characteristic values, which indicated that these were not suitable characteristic species for syntaxon F. In conclusion, two highly suitable negatively characteristic species were determined; however, suitably characteristic species were not found in syntaxon F.

**Table 2.** The first 5 high values of species for each syntaxon are listed with the species code (sc) and characteristic value (cv).

| Syntaxon | A | | B | | C | | D | | E | | F | |
|---|---|---|---|---|---|---|---|---|---|---|---|---|
| **Species** | **sc** | **cv** | **sc** | **cv** | **sc** | **cv** | **sc** | **cv** | **sc** | **cv** | **sc** | **cv** |
| 1 | GORDAX | 0.16 | ILEXGO | 1.03 | MELISQ | 0.23 | PSYCRU | 0.51 | HELIFO | 0.97 | CINNSU | 0.55 |
| 2 | ILEXLO | 0.16 | SYZYBU | 1.02 | ENGERO | 0.18 | SCHEOC | 0.48 | STYRSU | 0.69 | ORMOFO | 0.54 |
| 3 | MELACA | 0.15 | RANDCO | 1.02 | BLASCO | 0.18 | LITSAC | 0.43 | BLASCO | 0.22 | FICUFI | 0.15 |
| 4 | SCHISU | 0.15 | EUONLA | 0.57 | HELIRE | 0.18 | MICHCO | 0.26 | BEILER | 0.13 | SAURTR | 0.15 |
| 5 | ILEXMI | 0.06 | RHAPIN | 0.23 | TRICDU | 0.17 | HELICO | 0.25 | CLERCY | 0.12 | ARDIQU | 0.13 |

We implemented bipartite modularity by applying Newman's modularity measure in a weighted web to compute modules and their information in the network [33,40–42]. Consequently (Figure 6), the bipartite network can be grouped into five modules with the corresponding species group, also called the group of dominant characteristic species, in which syntaxa E and F share a similar pattern of species group in one module. The corresponding dominant species for syntaxon A happened to be its characteristic species, GORDAX. The dominant species for syntaxon B are RANDCO, SYZYBU, and EUONLA (*Euonymus laxiflorus* Champion ex Bentham), in which RANDCO and SYZYBU are the species with highly characteristic values. The dominant species for syntaxon C are DIOSMO (*Diospyros morrisiana* Hance.), HELIRE (*Helicia rengetiensis* Masam.), RANDCO, and BLASCO (*Blastus cochinchinensis* Lour.), in which BLASCO is not included in its species group. For syntaxon D, the dominant species are TRICDU (*Tricalysia dubia* (Lindl.) Ohwi), ARDIQU (*Ardisia quinquegona* Blume), CRYPCH (*Cryptocarya chinensis* (Hance) Hemsl.), MALLPA (*Mallotus paniculatus* Lam. Muell.-Arg.), PSYCRU, and SCHEOC. The syntaxa E and F have the same dominant species, BLASCO and HELIFO, but they are not included in the species group of F.

The PAC [43–45] of each syntaxa pair was calculated using species composition and visualized in the PAC network as a circular graph in which syntaxa were represented by circles and shared tree species were represented by connecting lines [33,46,47]. The results (Figure 7 and Table 3) revealed that syntaxon B had the highest PAC (0.54) within the syntaxon (i.e., B vs. B). In comparison with syntaxon B, the PACs of syntaxon A, C, D, E, and F were 0.24, 0.31, 0.25, 0.17, and 0.09, respectively. Syntaxon A exhibited a high PAC within the syntaxon, and syntaxon F exhibited the smallest PAC among other syntaxa. Syntaxon C exhibited the lowest PAC within the syntaxon. This study revealed the PAC network of syntaxa in the Lienhuachih forest area for the first time. As shown in Figures 3–7 and Table 3, the species composition of syntaxon A was unique; hence, the PAC within the community was the highest. The species compositions of the dominant species of syntaxon B and the other syntaxa were similar; consequently, the PAC relationship outside the syntaxon was the highest. Although syntaxon C exhibited relatively high species richness, no species was distinctly dominant; consequently, the PAC inside and outside the syntaxon

was very low. Figures 3 and 4 show that syntaxon C was also the succession prototype of all of the syntaxa in the area. Syntaxa D and E exhibited similar PAC structures, which showed that the common species structure of these two communities was similar. For syntaxon F, the external PAC with E was higher than that with others; this indicated the species composition structure and common species were similar between syntaxa E and F. These results indicate that PAC plays a crucial role in structuring community [46] in the forest of the Lienhuachih area, but the results do not clarify how the apparent competition govern the spatial distribution, species composition, biodiversity structure, and the succession of syntaxa. Additional studies are necessary to answer these questions.

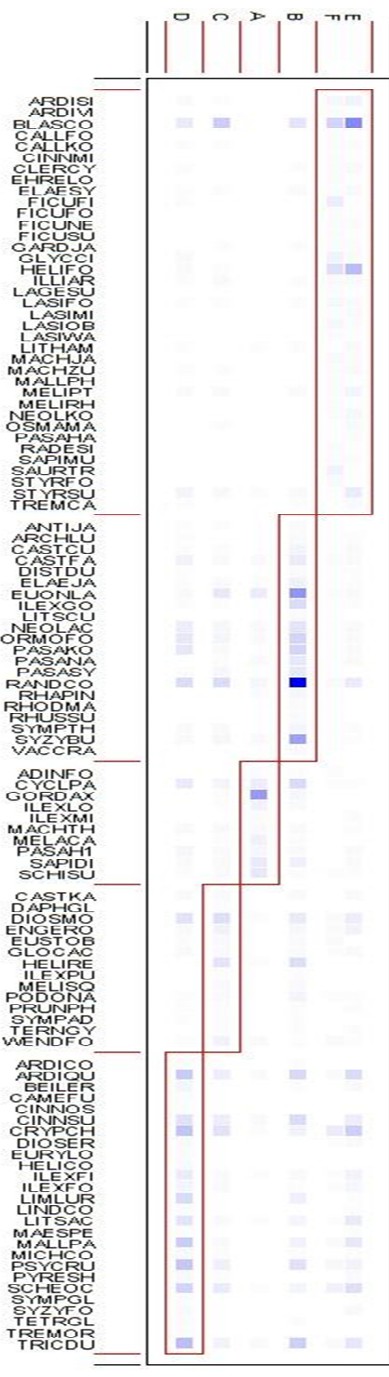

**Figure 6.** Modularity plot shows 5 modules with the abundance of the corresponding species group by the blue-scale.

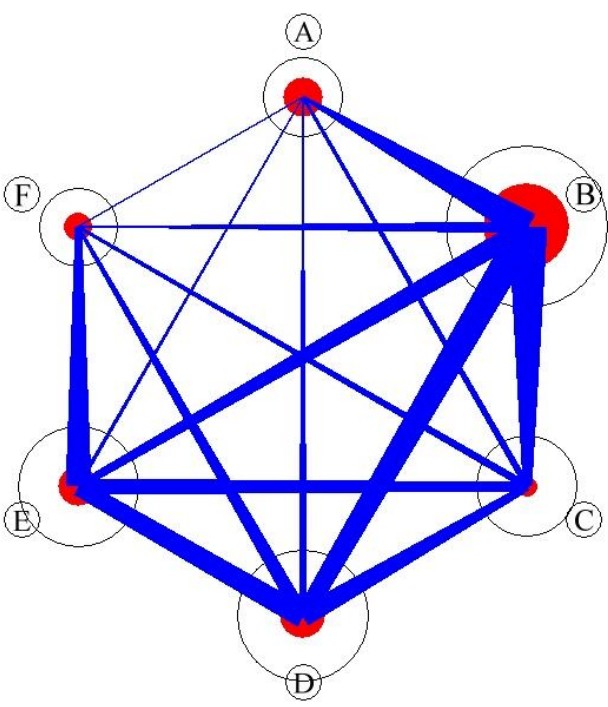

**Figure 7.** Quantitative circular graph of PAC network in the forest community of the study site. The circle's size is proportional to the species' abundance in the syntaxon; the red circles represent the apparent competition within syntaxon, and blue line presents the apparent competition among the syntaxa.

**Table 3.** PAC values in the network of syntaxa.

| Syntaxon | A | B | C | D | E | F |
|---|---|---|---|---|---|---|
| A | 0.52 | 0.24 | 0.09 | 0.09 | 0.05 | 0.01 |
| B | 0.06 | 0.54 | 0.12 | 0.17 | 0.09 | 0.02 |
| C | 0.05 | 0.31 | 0.19 | 0.2 | 0.18 | 0.06 |
| D | 0.03 | 0.25 | 0.11 | 0.36 | 0.19 | 0.05 |
| E | 0.02 | 0.17 | 0.13 | 0.22 | 0.34 | 0.12 |
| F | 0.02 | 0.09 | 0.1 | 0.15 | 0.28 | 0.36 |

Figure 8 shows the algorithm results from using the classification and regression tree of the dominant/characteristic species. The result demonstrated that RANDCO was used first to distinguish syntaxon B from the other species, in which syntaxon B was recorded more/equal to 53 and others were less than 53 RANDCO individuals. Then, PSYCRU helped distinguish syntaxon D (more/equal to 14) from the others (less than 14); HELIFO distinguished syntaxon A, C (less than 6) from the E, F (more/equal to 6), in which GORDAX was used to distinguish syntaxa A (more/equal to 30) from D (less than 30), and TRICDU was used to distinguish syntaxa E (more/equal to 8) from F (less than 8). In this study, 28%, 20%, 20%, 16%, 12%, and 4% of the quadrats belong to syntaxon B, C, D, E, F, and A, respectively. The rate of classification accuracy of the key to the six syntaxa was 100%.

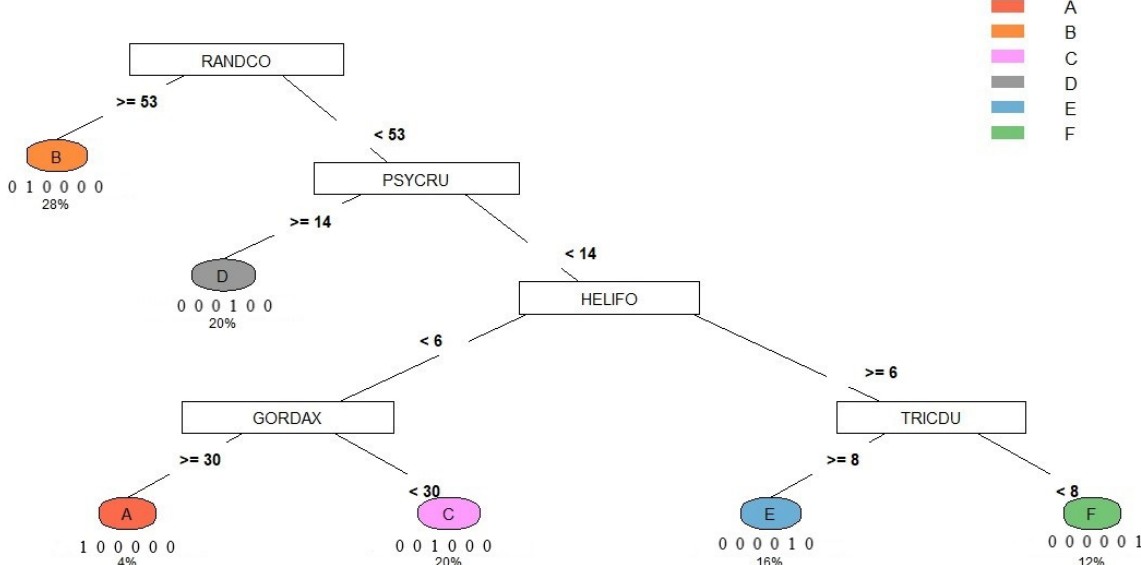

**Figure 8.** The key to syntaxa in the Lienhuachih area of Taiwan. The numbers below each syntaxon were the rate of classification accuracy and the rate of quadrats that belong to the syntaxon, respectively.

## 4. Discussion

Vegetation surveys have been conducted using small dispersed sample plots for several decades in Taiwan. Since 1990, to understand dynamic changes in forests, several large-forest dynamics plots, such as Nanjenshan, Nantzuhsienhsi, Fushan, and Lienhuachih, have been established [9]. We selected the Lienhuachih area as our study site because of its numerous advantages, such as having been extensively investigated for over 10 years, being exposed to low levels of human disturbance, exhibiting high biodiversity, and possessing a large cover of primeval forest area. Many of these are the characteristics of ideal study sites [54]. Consequently, many vegetation studies have been conducted at this experimental site in the past decade; however, none of these investigations have involved social networks. The social relationships among forest communities in this area have not been elucidated. Most vegetation analysts over the past few decades have preferred to conduct two-way indicator species analysis (TWINSPAN) or COCKTAIL [55]. Although the two methods yielded more objective results than traditional methods, they provided different results when applied to the same data set. Therefore, vegetation researchers urgently need to find more useful methods for solving the many problems in the field of vegetation ecology [56,57]. Accordingly, in this study, a combination of numerical taxonomy and network analysis techniques was used to study the forest community in the Lienhuachih area. Furthermore, the feasibility of applying these methods to vegetation analysis was examined. The results show that these methods have their own functions and are complementary to each other. The module analysis method could be used to identify modules and the dominant characteristic species group; hence, it is somewhat similar to the COCKTAIL method [58]. However, a starting species or species group must be preselected while using COCKTAIL. This pre-selection, to some extent, determines the final composition of the species group [57,58]. These problems have been reported by many vegetation ecologists. The results of the present study revealed the presence of six patterns or syntaxa in the forest community in the study site of the Lienhuachih area. The syntaxa E and F were similar; consequently, the module analysis and cluster analysis yielded different results. Figure 5 shows that the two syntaxa differed in rare species, while the common species were similar. Therefore, cluster analysis, which focus on overall similarities, and modular analysis, which focuses on dominant species, yielded different results. In a study by [9], four plant communities were identified and represented using dominant and indicator species based on TWINSPAN at the study site of the Lienhuachih

area. The communities were as follows: Type I, *Pasania nantoensis—Randia cochinchinensis*; Type II, *Mallotus paniculatus—Engelhardtia roxburghiana*; Type III, *Diospyros morrisiana—Cryptocarya chinensis*; Type IV, *Machilus japonica* var. *kusanoi—Helicia formosana*. Among the aforementioned communities, no community type corresponded with any syntaxa in the present study, and the four types were also highly different from those identified in the present study. Type I is most likely to correspond to syntaxon B in this study because the *Randia cochinchinensis* was the most-dominant characteristic species of pattern B, but *Pasania nantoensis* was the fourteenth-most dominant characteristic species of pattern B (Figure 6). Type II may correspond to syntaxon C or D, because *Mallotus paniculatus* was the fifth-most dominant characteristic species of syntaxon D, and *Engelhardtia roxburghiana* was the tenth-most dominant characteristic species of syntaxon C, but neither of these two species was the most-dominant characteristic species of syntaxa C and D. Type III may also correspond to syntaxa C or D, because *Diospyros morrisiana* was the most-dominant characteristic species of syntaxon C, and *Cryptocarya chinensis* was the second-most dominant characteristic species of syntaxon D. Type IV probably corresponds to the syntaxa E and F in the present study, because both *M. japonica* var. *kusanoi* and *Helicia formosana* were among the dominant characteristic species of syntaxa E and F, but neither was the most-dominant characteristic species. *Machilus japonica* var. *kusanoi* was the thirty-seventh-most dominant characteristic species of syntaxon E and the fifth of syntaxon F. *Helicia formosana* was the second-most dominant characteristic species of syntaxa E and F. Therefore, the results of this study differed substantially from those of a previous study conducted using TWINSPAN.

The detected community patterns or syntaxa in the present study were consistent with the definitions of an association in all aspects [59,60]. Therefore, the next question was the nomenclature of these associations. The nomenclature of syntaxa has evolved over a long time with many controversies [4,57,60]; for example, the scientists of the Clements school prefer to use nomenclature coming from associations defined by dominant species, whereas scientists of the Floristic school [3] prefer nomenclature based on the characteristic species, and they proposed the International Code of Phytosociological Nomenclature (ICPN) [61]. Scientists from many countries, such as the United Kingdom [62], China [58], Taiwan [63,64], and the United States [59], have proposed principles of vegetation classification and nomenclature. Although numerical classification has enabled many useful statistical methods in recent years, a nomenclature system supporting numerical methodology has not been proposed, and the nomenclature of plant communities has not yet been unified globally [56,58,65]. Therefore, we referred to the literature and case studies [13,55,58,61,66–72] and proposed in this study a binomial nomenclature system for numerical syntaxonomy that meets the requirements of statistical analysis results. This nomenclature system, the Mathematic Code of Syntaxonomic Nomenclature (MCSN), adopts the most-dominant characteristic species in a module as the generic name and the most-characteristic species as the specific epithet for an association. In the middle of the generic name and the specific epithet is a mathematical symbol, plus or minus, which represents the fact that the association is named by the MCSN. The PAM algorithm is used to designate the nomenclature type relevé of an association. According to the MCSN, association A can be named Assoc. *Gordonia axillaris + Gordonia axillaris* Tung-Yu Hsieh et al., 2022, and the nomenclature type relevé is x15y20, which indicates both that the association is published by Hsieh et al. (2022) and that *Gordonia axillaris* is the most-dominant characteristic species. Association B can be named Assoc. *Randia cochinchinensis + Ilex goshiensis* Tung-Yu Hsieh et al., 2022, the nomenclature type relevé is x15y5, which represents an association with *R. cochinchinensis* as the most-dominant characteristic species and *Ilex goshiensis* as the most-characteristic species. Association C can be named Assoc. *Diospyros morrisiana + Meliosma squamulata* Tung-Yu Hsieh et al., 2022, and the type relevé is x20y20. Association D can be named Assoc. *Tricalysia dubia + Psychotria rubra* Tung-Yu Hsieh et al., 2022, and the type relevé is x5y10. Association E can be named Assoc. *Blastus cochinchinensis + Helicia formosana* Tung-Yu Hsieh et al., 2022, and the type relevé is x10y0. Association F can be named Assoc. *Blastus cochinchinensis-Cinnamomum subavenium* Tung-Yu Hsieh et al., 2022, and the middle symbol

of the scientific name is a minus, which indicates that the most-dominant characteristic species is *Blastus cochinchinensis* and that *Cinnamomum subavenium* is the most negatively characteristic species of the association, and the type is x5y20. The characteristic species data of nomenclature types relevés A–F are presented in Table 4. In total, five genera and six species of associations were identified in the study site of the Lienhuachih area. Among the associations, E and F are associations of the same genus. Syntaxa belonging to the higher or lower ranks that are not mentioned in this study can also be applied to this nomenclature system. In addition, the MCSN retains valid publications of authors and year and the codes of priority in the ICPN and discards the suffix changes and tautonymous names in the list of scientific names to improve compatibility with the properties of the syntaxon. Because binomial nomenclature and this form of naming and presenting associations have been used by many biologists in previous studies on vegetation ecology, we are only redefining them now. The prefix word "Assoc." identifies the rank of a syntaxon; hence, the suffix of the scientific name need not be changed to indicate its rank, making the MCSN easier to understand. If some scholars cannot accept such a nomenclature system, they can also rename the syntaxon according to their own preference. For example, according to the Braun-Blanquet approach (ICPN), these associations would be renamed as follows:

**Table 4.** Characteristic species data of the nomenclature type relevés.

| Type Releve' | ARD IQU | BLA SCO | CIN NSU | CRY PCH | DIO SMO | EUO NLA | FIC UFI | GOR DAX | HEL IFO | HEL IRE | ILE XLO | ILE XGO | MAL LPA | MEL ACA | MEL ISQ | ORM OFO | PSY CRU | RAN DCO | SAU RTR | SCH EOC | SYZ YBU | TRI CDU | Syntaxon Code |
|---|---|---|---|---|---|---|---|---|---|---|---|---|---|---|---|---|---|---|---|---|---|---|---|
| x15y20 | 3 | 0 | 2 | 1 | 1 | 10 | 0 | 57 | 0 | 1 | 6 | 0 | 0 | 7 | 0 | 3 | 0 | 4 | 0 | 1 | 3 | 2 | A |
| x15y5 | 31 | 0 | 7 | 0 | 20 | 62 | 0 | 8 | 0 | 30 | 0 | 25 | 21 | 3 | 0 | 3 | 2 | 318 | 0 | 2 | 52 | 33 | B |
| x20y20 | 13 | 30 | 6 | 6 | 21 | 12 | 0 | 3 | 0 | 3 | 0 | 0 | 1 | 0 | 1 | 9 | 1 | 21 | 0 | 12 | 2 | 4 | C |
| x5y10 | 49 | 14 | 21 | 14 | 9 | 0 | 0 | 0 | 3 | 0 | 0 | 0 | 3 | 0 | 0 | 9 | 18 | 33 | 0 | 24 | 2 | 61 | D |
| x10y0 | 17 | 121 | 6 | 16 | 10 | 0 | 0 | 0 | 28 | 0 | 0 | 0 | 2 | 0 | 0 | 1 | 8 | 4 | 0 | 28 | 0 | 10 | E |
| x5y20 | 2 | 25 | 0 | 7 | 2 | 0 | 6 | 0 | 10 | 0 | 0 | 0 | 3 | 0 | 0 | 0 | 1 | 0 | 1 | 4 | 0 | 0 | F |

*Syntaxonomical Synopsis*

A    Assoc. *Gordonietum axillare* Tung-Yu Hsieh et al., 2022 ass. nova hoc loco

Nomenclature type releve': x15y20 (holotypus hoc loco designatus).

B    Assoc. *Randio cochinchinensis-Iletum goshiensis* Tung-Yu Hsieh et al., 2022, ass. nova hoc loco

Nomenclature type releve': x15y5 (holotypus hoc loco designatus).

C    Assoc. *Diospyro morrisianae-Meliosmetum squamulatae* Tung-Yu Hsieh et al., 2022, ass. nova hoc loco

Nomenclature type releve´: x20y20 (holotypus hoc loco designatus).

D    Assoc. *Tricalysio dubiae-Psychotretum rubrae* Tung-Yu Hsieh et al., 2022, ass. nova hoc loco

Nomenclature type releve´: x5y10 (holotypus hoc loco designatus).

E    Assoc. *Blasto cochinchinensis-Helicetum formosanae* Tung-Yu Hsieh et al., 2022, ass. nova hoc loco

Nomenclature type releve´: x10y0 (holotypus hoc loco designatus).

F    Assoc. *Blasto cochinchinensis-Cinnamometum subaveniae* Tung-Yu Hsieh et al., 2022, ass. nova hoc loco

Nomenclature type releve': x5y20 (holotypus hoc loco designatus).

In general, the names of associations A–E were sufficiently compatible between the ICPN and MCSN. Only the properties of the association F cannot be adequately described by the ICPN. However, when named as follows, the characteristics of the association F were clearer.

Assoc. *Blasto cochinchinensis-Ficetum fistulosae* Tung-Yu Hsieh et al., 2022, ass. nova hoc loco
Nomenclature type releve': x5y20 (holotypus hoc loco designatus).

In this study, a classification and regression tree was adopted to replace the manmade key. Compared with the conventional key, the classification and regression tree had great processing power and could carry out more precise algorithms for many numerical characteristics that are excellent in classification. For large-scale and complicated data, this advantage is especially evident. Another advantage of the classification and regression tree is its great flexibility of application. Not only can it offer professional biologists a method of classification through an observable, comprehensible, and accessible presentation like the conventional key, but it can also be used as a computer algorithm program that accesses the database from the internet and the computer programs to offer ordinary people quick species identification online. This is what conventional keys find hard to achieve.

Taxonomy and vegetation classification have a similar theoretical background, so the problems they encountered in the development process were also very similar. Early research on taxonomy and vegetation classification was often questioned because the research was subjective and lacked the use of objective methods. Although numerical taxonomy methods have improved some of the issues, many problems remain unsolved, such as type designation, the discovery of new taxa/syntaxa, revision of scientific names, and the production of a key, these important works in taxonomy/vegetation classification still lack objective analysis methods. After many years of case studies, the authors have gradually proposed appropriate numerical analysis methods for the above problems [13,72] and call them "taxonomy modeling" or "classifier modeling".

## 5. Conclusions

In recent years, statistical technology has made significant progress in taxonomy and social network research. However, the methods from social network analysis are still rarely used in vegetation research. Therefore, this study takes the plots of the Lienhuachih area as a case study and tries to combine some new methods from different disciplines into vegetation ecology.

In this study, the results of the heat map and gap analysis showed that a total of 108 tree species of six different syntaxa were investigated in the lotus pond area, and A–F codes were assigned to six different syntaxa. Through clustplot and a minimum spanning tree, it can be found that B is the most common syntaxa in the area, A is the rarest, and C is the succession prototype of the local community patterns; the Renyi index shows that D is the highest diversity syntaxon, and A is the lowest one; the results of Newman's modularity measure showed that there are five different modules in the Lienhuachih area. The results of this study showed that the numerical taxonomy methods are good at discriminating the species structure, and the modular method derived from social network analysis is good at distinguishing the generic structure. These two methods complement each other and apply to the analysis of the vegetation ecology; they could solve the rank problems that are not easily discernible in past vegetation research. The network of PAC showed that the most common syntaxon B in Lienhuachih had the greatest PAC, while C had the least PAC among other syntaxa. This indicates that, when B is adjacent to other syntaxa, they will have a larger PAC and be more susceptible to the common influence factor.

In the past, in the syntaxonomic nomenclature system, as with the taxonomic nomenclature system of species, the lack of objective analytical techniques often led to many inconsistencies in the nomenclature of the syntaxa in different studies. Therefore, this study redefines some nomenclature codes based on the properties of numerical analysis methods. The Mathematic Code of Syntaxonomic Nomenclature (MCSN) is proposed for the syntaxon naming system, but this does not mean that the traditional naming system must be abandoned. We just proposed a naming system that is more in line with the numerical analysis methods.

From the case study of Lienhuachih, we can find that these new methods, which are derived from numerical taxonomy and social network analysis, can be well combined with the research of vegetation analysis. This study, combined with several of our relevant published research cases, has solved the problem of lacking objective analysis methods

in many key works of these two disciplines and has completed, unifying taxonomy and vegetation classification in methodology. In the future, these methods can be used in taxa/syntaxa classification but also have great application potential in the research of germplasm resources, such as the discovery of new cultivars or the identification of cultivars. The authors believe that there will be more new methods in different fields that will be applied to the field of vegetation ecology/taxonomy in the future. This research is only the beginning of the application of social network methods in these two fields.

**Author Contributions:** T.-Y.H. contributed significantly to data analysis and wrote the manuscript; C.-R.C. contributed to the conception of the study; C.-J.Y. performed the vegetation survey; F.L. helped perform the analysis with constructive discussions. All authors have read and agreed to the published version of the manuscript.

**Funding:** This study was supported by the Chinese Academy of Sciences (Grant no.: 2013 TW2SA 0003, 2015TW1SA0001), the Forestry Bureau, Council of Agriculture, Executive Yuan (Grant no.: 101 agriculture-13.5.4-forestry-e1), and the National Science Council (Grant no.: NSC 102-2313-B-002-038).

**Institutional Review Board Statement:** "Not applicable" for studies not involving humans or animals.

**Data Availability Statement:** The data presented in this study are available on request from the corresponding author.

**Acknowledgments:** We are thankful to the Forestry Research Institute for providing the data on the Lienhuachih forest dynamic samples and the Forestry Bureau for providing the national flora map.

**Conflicts of Interest:** The authors declare no conflict of interest.

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
