# Peer review of "Numerical Ecology and Social Network Analysis of the Forest Community in the Lienhuachih Area of Taiwan"

_diversity, doi:10.3390/d15010060_

Round 1

Reviewer 1 Report

This reviewer has been previously unaware of the possibility of applying social network methods to vegetation ecology. The paper appears innovative, the study is carefully designed and professionally reported. Minor polishing of the English language appears to be needed.

This reviewer does not have much experience with the methods used in the study and consequently is unable to provide more specific comments.

Author Response

1. Minor polishing of the English language appears to be needed.

Response: The manuscript has been revised by Wallace Academic Editing.

Reviewer 2 Report

Interdisciplinary studies are of high importance and are needed to solve complex environment problems.  The present study is well-designed, but needs to address a number of issues:

The introduction part needs to be rewritten. In the introduction, the current problem statement needs to be done. Why your study is needed? What did other researchers say? What is the gap of knowledge? What is the state of art of your study? 

Clearly state the aim of the research and the objectives of the study.

The Humboldt - please add the reference.

Line 51-60 to the methods, as it is the description of the study area.

Line 61-70 to Study areas

Line 201-219 please use Italic for the Latin names of plants

Fix the references, N 12, 15, 21, 23, 25, 26, 28, 46, 63

Author Response

  1. The introduction part needs to be rewritten. In the introduction, the current problem statement needs to be done. Why your study is needed? What did other researchers say? What is the gap of knowledge? What is the state of art of your study?

Response: We have rewritten the introduction part according to the comments of the reviewer’s comment, see line 33-86, please.

  1. Clearly state the aim of the research and the objectives of the study.

Response: We state it on line 81-86.

  1. The Humboldt - please add the reference.

Response: We’ve added it, see lines 35 and 441, please.

  1. Line 51-60 to the methods, as it is the description of the study area.

Response: We’ve changed it to study area, see line 89-95, please.

  1. Line 61-70 to Study areas

Response: It seems to belong to the discussion, we’ve changed it to line 304-314.

  1. Line 201-219 please use Italic for the Latin names of plants

Response: Done, see line 233-269, please.

Fix the references, N 12, 15, 21, 23, 25, 26, 28, 46, 63

Response: Done, see lines 469, 473, 486, 489, 492, 495, 499, 529, 559, respectively.

Reviewer 3 Report

This is a very interesting study focused on the use of more advanced statistical methods and supported by a number of references. I have only a few comments on the manuscript:

L24-25 and the relevant part of the results: Despite I am impressed with the analysis, I think it is too bold aimed to propose a new Syntaxonomic Nomenclature based on just 25 samples – a much larger dataset should be behind this type of work. Reconsider deleting this part of the discussion.

L68: „beenelucidated“ – been elucidated

L77: Upper index for 2 in m2

L83: I think it would be more appropriate to state that you used "multiple methods" without the evaluative adjective "useful"

Table 1: Information about syntaxa always follows taxa information, except for the first two rows. Please unite with the rest of the table.

L118-L120: These statements should still be part of the methodology or already part of the discussion. It doesn't belong to the results.

L129-L133: Although it is clear from Figure 1 and Figure 2 why you chose to select six clusters, some more technical description of how this was achieved would be appropriate. For example, based on the fact that only between 5-6 or between 22-23 clusters there is a bigger difference in the average of clusters than the standard deviations of clusters 5-6 or 22-23?

Figure 3 but also others: Many plots are of poor quality. This is sufficient for peer review, but keep in mind to supply higher quality plots for publication.

3.2. Social networks analysis: Some of the statements would deserve a more detailed explanation of their meaning from the point of view of ecology. E.g.: what does it mean that „syntaxa were more vulnerable than the taxa“? or „higher values of togetherness of the syntaxa indicated that the taxa exhibited disaggregation than the syntax? The values of robustness indicated that the syntaxa were more vulnerable to secondary extinction? It would also be appropriate to state specific values here again, as in the case of the other characteristics mentioned in the given paragraph.

L201-L222 and L226-L241: Taxa mentioned in these paragraphs should be in italics

L265-L268: Why does this sentence end with a question mark? It doesn't make sense grammatically

L294-L295: „Therefore, cluster analysis, which focus on overall similarities, 294 and modular analysis, which focuses on dominant species“ … analysis focuses/analyses focus, correct

L318-L319: What does "this analysis was more consistent with raw dataset" mean - how is this consistency measured?

L364-L367: Because binomial nomenclature and this form of naming and presenting associations have been used by many biologists in previous studies on vegetation ecology, we are only redefining them now, to facilitate public acceptance of the MCSN should be more easily accepted by the general public.“ … The end of the sentence does not make grammatical sense.

Author Response

  1. L24-25 and the relevant part of the results: Despite I am impressed with the analysis, I think it is too bold aimed to propose a new Syntaxonomic Nomenclature based on just 25 samples – a much larger dataset should be behind this type of work. Reconsider deleting this part of the discussion.

Response: In order to respect the reviewers' opinions, we temporarily delete this part, but we still hope that the reviewer can allow us to recovery it after consideration. For two reasons, first, all naming systems are proposed based on demand, regardless of the number of samples. Second, although there are many naming systems, none of them are proposed based on the requirements of numerical analysis. Therefore, we need a naming system that can be matched with numerical analysis.

  1. L68: „beenelucidated“ – been elucidated

Response: Done, see line 314.

  1. L77: Upper index for 2 in m2

Response: Done, see line 100.

  1. L83: I think it would be more appropriate to state that you used "multiple methods" without the evaluative adjective "useful"

Response: Done, see line 114.

  1. Table 1: Information about syntaxa always follows taxa information, except for the first two rows. Please unite with the rest of the table.

Response: Table 1 reveals the network structure indicators. The syntaxa is the nodes of a web, not for any specific syntaxon here.

  1. L118-L120: These statements should still be part of the methodology or already part of the discussion. It doesn't belong to the results.

Response: We’v changed it to line 119-121.

  1. L129-L133: Although it is clear from Figure 1 and Figure 2 why you chose to select six clusters, some more technical description of how this was achieved would be appropriate. For example, based on the fact that only between 5-6 or between 22-23 clusters there is a bigger difference in the average of clusters than the standard deviations of clusters 5-6 or 22-23?

Response: Yes, that it. We’ve added the precious comments on line 154-156.

  1. Figure 3 but also others: Many plots are of poor quality. This is sufficient for peer review, but keep in mind to supply higher quality plots for publication.

Response: We have reproduced several new pictures for publication, see line 163, 165, 184, 188, 202, 272, and 297.

  1. 2. Social networks analysis: Some of the statements would deserve a more detailed explanation of their meaning from the point of view of ecology. E.g.: what does it mean that „syntaxa were more vulnerable than the taxa“? or „higher values of togetherness of the syntaxa indicated that the taxa exhibited disaggregation than the syntax? The values of robustness indicated that the syntaxa were more vulnerable to secondary extinction? It would also be appropriate to state specific values here again, as in the case of the other characteristics mentioned in the given paragraph.

Response: We are glad the reviewer has an interest in social networks. However, the literature we cited has explained these questions in detail. We suggest that readers read the literature because it is beyond our ability to explain these complex problems in a short paragraph, though we’ve tried our best as possible as we can in the past several days. We welcome the reviewer can provide more specific comments to help us revise this part.

  1. L201-L222 and L226-L241: Taxa mentioned in these paragraphs should be in italics

Response: Done, see line 233-269, please.

  1. L265-L268: Why does this sentence end with a question mark? It doesn't make sense grammatically

Response: We’ve delete the mark, see line 296, please.

  1. L294-L295: „Therefore, cluster analysis, which focus on overall similarities, 294 and modular analysis, which focuses on dominant species“ … analysis focuses/analyses focus, correct

Response: Yes, it is. We welcome the reviewer could provide more specific comments to make the meaning of this paragraph clear.

  1. L318-L319: What does "this analysis was more consistent with raw dataset" mean - how is this consistency measured?

Response: We’ve deleted the paragraph because we didn’t measure it.

  1. L364-L367: Because binomial nomenclature and this form of naming and presenting associations have been used by many biologists in previous studies on vegetation ecology, we are only redefining them now, to facilitate public acceptance of the MCSN should be more easily accepted by the general public.“ … The end of the sentence does not make grammatical sense.

Response: This part has been deleted now.

Round 2

Reviewer 2 Report

Please add the paragraph where you discuss the limitations of your study. What are the main gaps in your research? 

If another scientist would like to repeat your study, will it be possible to use your methods description? If not, please improve the methods part. 

In the conclusions part please highlight what your research has added to the body of knowledge in taxonomy and social network research.

Author Response

Dear Milica Čudić and Reviewers,

Thanks very much for taking the time to review this manuscript. We appreciate all your comments and suggestions! Please find our itemized responses in the revised manuscript.

Sincerely,

Tungyu Hsieh

Response to reviewer’s comments

  1. Please add the paragraph where you discuss the limitations of your study. What are the main gaps in your research?

Response: We've added it in lines 451-464.

  1. If another scientist would like to repeat your study, will it be possible to use your methods description? If not, please improve the methods part.

Response: We think that if another scientist has R language programming ability and refers to the relevant literature we cited, it is no problem to repeat our analysis methods. If necessary, anyone can also ask the authors for the R code through email.

  1. In the conclusions part please highlight what your research has added to the body of knowledge in taxonomy and social network research.

Response: We've added it in lines 497-505.
